# Savinin Triggers Programmed Cell Death of Ray Parenchyma Cells in Heartwood Formation of *Taiwania cryptomerioides* Hayata

**DOI:** 10.3390/plants12173031

**Published:** 2023-08-23

**Authors:** Nai-Wen Tsao, Ying-Hsuan Sun, Fang-Hua Chu, Shih-Chang Chien, Sheng-Yang Wang

**Affiliations:** 1Special Crop and Metabolome Discipline Cluster, Academy Circle Economy, National Chung Hsing University, Taichung City 402202, Taiwan; nwt1228@dragon.nchu.edu.tw; 2Department of Forestry, National Chung-Hsing University, Taichung City 402202, Taiwan; yhsun@nchu.edu.tw; 3School of Forestry and Resource Conservation, National Taiwan University, Taipei City 106319, Taiwan; fhchu@ntu.edu.tw; 4Experimental Forest Management Office, National Chung-Hsing University, Taichung City 402202, Taiwan; scchien@dragon.nchu.edu.tw; 5Agricultural Biotechnology Research Center, Academia Sinica, Taipei City 115201, Taiwan

**Keywords:** *Taiwania cryptomerioides* Hayata, heartwood formation, lignan biosynthesis, programmed cell death, savinin

## Abstract

The purpose of this study was to investigate the relationship between lignan biosynthesis and programmed cell death (PCD) of ray parenchyma cells during the heartwood formation of Taiwania (*Taiwania cryptomerioides* Hayata). Since the PCD of ray parenchyma cells and the synthesis of lignans are the two main processes involved in the formation of heartwood, both of which need to be completed through gene regulation. Based on the results of genomics and bioinformatics analysis, that the PCD of tracheids are induced by genotoxic, and the PCD of ray parenchyma cells is induced by biological factors, such as fungi, bacteria, and viruses, which could induce oxidative stress. According to the results of time−of−flight secondary ion mass spectrometry (ToF−SIMS) analysis, lignans are produced in ray parenchyma cells, and the accumulation of savinin and its downstream lignans might be the cause of PCD in ray parenchyma cells. An in vitro experiment further confirmed that the accumulation of savinin could cause protoplasts of Taiwania’s xylem to produce taiwanin A, which is the marker of heartwood formation in Taiwania. Resulting in an increase in reactive oxygen species (ROS) content, which could induce oxidative stress in ray parenchyma cells and potentially lead to PCD. Based on these findings, we conclude that accumulation of savinin could be induced PCD of ray parenchyma cells in heartwood formation in Taiwania.

## 1. Introduction

Trees are the answers to conquer the environmental challenges, such as climate warming, shortage of energy and resources, reduction of tropical rain forests, desertification of land, and endangered species of wild animals and plants. In addition to fixing carbon to slow down CO_2_ content in the atmosphere, trees provide wildlife habitat, and conserve water. It was the raw materials for the paper industry, interior decoration, furniture, and even for pharmaceuticals. Heartwood is a large part of the wood, and its extractives contained in it are closely related to the properties of the wood, such as durability, color, and odor [1]. The formation mechanism of heartwood has always been of interest and an important research topic for researchers [2]. The International Association of Wood Anatomists (IAWA) subdivided wood into the sapwood, and heartwood in 1964 and further defined that sapwood locates on the outside of the wood of the tree, with living cells and storage of primary metabolites (such as starch) [3]. The heartwood locates in the inner layer of the tree’s wood, there are no living cells, and the primary metabolites convert into heartwood substances. The transition zone, located in the narrow zone between the sapwood and the heartwood, starts from the inner layer of the sapwood [4]. Only parenchyma cells are still alive in this area, and they will begin to change in color and characters. For conifers, living parenchyma cells in the sapwood account for about 5–35%, while broad-leaved trees account for 10–35% [5]. According to the gradual decrease of oxygen, starch, and lipid content from cambium to transition zone, it can be inferred that parenchyma cells in the sapwood migration transition zone is a gradual aging process. The starch and lipids will convert into heartwood extractives in parenchyma cells, the accumulation of the heartwood extractives will make the heartwood and sapwood have notable color changes [5]. Most woods are easy to distinguish between heartwood and sapwood based on their color changes, and heartwood also has a darker color due to the accumulation of a large number of extractives [6,7,8].

The change of the sapwood to the heartwood was also characterized by the change of the chemical composition such as extractives [9,10,11,12,13,14,15,16,17,18]. Based on the distribution patterns of extractives, the heartwood formation was classified into two types [19]. Type I heartwood formation, i.e., Robinia—type heartwood formation, where the accumulation of phenolic extractives starts in the transition zone. In this case, no phenolic precursors were found in the aging sapwood. Type II (Juglans—type) heartwood formation, where the phenolic precursors gradual accumulated centripetally with progressive aging of the sapwood tissues. The extractives that characterize the Type II heartwood were formed in the TZ either by de novo biosynthesis or secondary reactions (oxidation or hydrolysis) of precursor substances. However, our group found that the distribution patterns of extractive was different from either type in Taiwania (*Taiwania cryptomerioides* Hayata) tree species [2,20]. In the Taiwania trees, most lignan compounds are synthesized (secondary reactions) in sapwood [2,20], so we call it Type III (Taiwania—type) heartwood formation. Taiwania has excellent physical processing properties and is widely used in wood products. The phytochemistry of Taiwania have been studied since the 1930s. More than hundreds of compounds with bioactivities have been identified from different parts of Taiwania [1,2,21,22]. Taiwania is the appropriate material for studying heartwood formation, due to it can easily distinguish the heart wood, sapwood wood, and transition zone according to the color. This study combines genomics and bioinformatics analysis with studying the related regulatory genes of lignan biosynthesis and the inducing mechanism of programmed cell death (PCD) in parenchymal cells. A Time−of−Flight Secondary Ion Mass Spectrometer (ToF−SIMS) and an in vitro test system established using the protoplasts of the Taiwania were used to explore the relationship between lignans and PCD in ray parenchyma cells.

## 2. Results and Discussion

### 2.1. Types of Induced Programmed Cell Death

Heartwood is defined as the death of parenchyma cells, which means that heartwood formation can be considered as the process of parenchyma cell death; studies on tissue anatomy have confirmed this hypothesis [23,24,25]. Also, there are reports using in situ hybrid technology to stain dirigent protein, proving that parenchyma cells are the central part of lignan biosynthesis [26]. Taiwania caused the decline of starch content, the deposition of extractives, and the death of ray parenchyma cells [27]. Therefore, the regulation mechanism of PCD in ray parenchyma cells and the biosynthetic pathway of lignans will play a critical role in forming heartwood.

There are two stages of PCD during the wood formation of Taiwania. The first stage is the process of cambium entering the sapwood during differentiation. At this time, the tracheids are undergoing PCD. The second stage is the process in the transition zone. At this time, the xylem parenchyma cells undergo PCD. Olvera−Carrillo and his coworker identified four categories of PCD−related expression genes, developmentally related, biotic, osmotic, and genotoxic [28]. In this study, gene expression differences in differentiating xylem (DX), sapwood (SW), and transition zone (TZ) from Taiwania based on the expression genes of programmed cell death between two stages was analyzed according to Olvera−Carrillo’s classification (Figure 1). When the cambium transformed into sapwood during the differentiation, the genes related to genotoxicity−induced increased during the differentiation in the xylem. However, the gene expression level in sapwood decreased because most of the cell was dead. In addition, the expression levels of genes of other induced−type were not consistent (Figure 1a). Therefore, the PCD in secondary xylem of Taiwania belonged to a genotoxic−induced PCD in the first stage, which means the tracheids of Taiwania died through cytotoxicity in the first stage of PCD. On the other hand, when the sapwood transforms into the transition zone, most increased expression genes were related to the biological factor−induced type in the transition zone (Figure 1b). In contrast, the other induced gene expressions did not find irregular changes. Therefore, the second stage of PCD in the transition zone belong to biological factor−induced PCD. Most of the biological factor−induced PCD caused by pathogenic bacteria infection; others include cell aging, ultraviolet irradiation, oxidative stress, etc. [28]. The process of conifers’ heartwood formation would accumulate massive lignans. The lignans of Taiwania possessed potent cytotoxicity against cancer cells [1,29]. Therefore, the abundant lignans may induce programmed cell death of ray parenchyma cells in heartwood formation of Taiwania.

### 2.2. Positioning Analysis of Lignans in the Tangential Section of Taiwania

We have demonstrated the significant accumulation of six lignans (hinokinin, savinin, taiwanin A, taiwanin C, taiwanin E, and helioxanthin) within the transition zone of Taiwania [2,18]. Furthermore, existing literature highlights the potential association between the transformation of sapwood into heartwood and the generation of deleterious compounds [30]. Based on this premise, we postulated that the initiation of heartwood formation in Taiwania might be instigated by the programmed cell death (PCD) of ray parenchyma cells induced by lignans. In order to confirm whether the biosynthesis of lignans were related to the PCD of ray parenchyma cells, the ToF−SIMS was used for analyzing the position of lignans, and for the convenience of observation, the angles of images were adjusted to arrange ray parenchyma cells. The results showed in Figure 2. It was difficult to distinguish between ray parenchyma cells and tracheids from the total ion image distribution map of ToF−SIMS. However, the potassium ions (K^+^) exist in living wood cells, e.g., tracheids and parenchyma cells and related with the cell expansion in the development of the xylem [31,32,33,34,35]. Only parenchyma cells are still alive in the sapwood. Thus, ray parenchyma cells of Taiwania can be distinguished from the accumulation of K^+^. as shown in Figure 2, K^+^ could be accumulated in tangential section specimens of sapwood (samples 1–4) and slight in heartwood (sample 5), the results were constancy with the survival of ray parenchyma cells. Therefore, we could use it to map the ray parenchyma cells. The image distribution of calcium ions (Ca^2+^) shows that Ca^2+^ accumulates only in the front part of the sapwood (sample 1), but there is no evident accumulation of Ca^2+^ when moving to the heartwood, and Ca^2+^ gradually disappeared during sample 1 to 5. It has been mentioned that Ca^2+^ can affect the development of the xylem through PCD [32,36,37], and previous studies have pointed out that in the transition zone and heartwood, Ca^2+^ accumulation can be observed [11,38]. The ray parenchyma cells survive until the transition zone and begin to die in Taiwania [27]. The Ca^2+^ image distribution map result, which not accumulated in the PCD of Taiwania ray parenchyma cells.

We furthermore analyzed the lignans image distribution map in secondary xylem in Taiwania. Dividing the intensity of the signal by the total ion intensity to quantification (Appendix A). These six lignans were more abundant in the heartwood than sapwood. Especially, taiwanin A only appeared in the heartwood. These results are consistent with our previous study [2,20], and again confirm taiwanin A is produced by heartwood formation; in other word, taiwanin A could be a marker of heartwood formation. However, according to distribution map that hinokinin was accumulated in ray parenchyma cells of sapwood, but not concentrated in ray parenchyma cells in heartwood. Savinin, taiwanin C, and helioxanthin were accumulated in ray parenchyma cells, and even highly concentrated in the heartwood. In particular, savinin has a high concentration in ray parenchyma cells of sapwood (sample 3 and 4). Taiwanin A was only accumulated in ray parenchyma cells in sample 5 (heartwood).

The findings underscore direct lignan synthesis within ray parenchyma cells. Notably, among these lignans, savinin, taiwanin C, helioxanthin, and taiwanin A exhibited pronounced concentration levels in the heartwood (sample 5). When consulting the lignan biosynthetic pathway (Appendix A), it becomes evident that savinin functions as the precursor to taiwanin A, taiwanin C, and helioxanthin [2]. In accordance with Olvera−Carrillo’s classification, the induction of programmed cell death (PCD) in ray parenchyma cells was brought about by oxidative stress. Our investigations affirm higher oxidative stress levels within the transition zone and heartwood in comparison to sapwood (Appendix A). Furthermore, the accumulation of lignans within the ray parenchyma cells of Taiwania was observed, with savinin exhibiting the highest concentration amongst them. This particular compound holds the potential for conversion into taiwanin A, taiwanin C, and helioxanthin. As a result, the accrual of savinin could play a pivotal role in heartwood formation, triggering oxidative stress in ray parenchyma cells and thereby facilitating the significant synthesis of taiwanin A, taiwanin C, and helioxanthin.

### 2.3. Detection of Taiwanin A through In Vitro Assay

A protoplast system was conducted to verify that lignans are the signal for triggering PCD in this study. The initiation of PCD in ray parenchyma cells might represent the beginning of heartwood formation. Our previous study and the above results indicated that the content of lignans increased significantly, and ray parenchyma cells died immediately after entering the transition zone, where taiwanin A starts to be produced in Taiwania [2,20,27]. Thus, the detection of taiwanin A could be the index of heartwood formation. Table 1 shows the results of quantitative analysis by UHPLC−MS after 24 h of different lignans treatments in the protoplast cells culture system. The results revealed that the control group could produce small amounts of hinokinin and savinin. The savinin was synthesized after calcium chloride treatment for 24 h in protoplast cells. However, no lignans were found during hydrogen peroxide treatment. On the other hand, the savinin, taiwanin C and helioxanthin could be detected after 24 h of treatment with hinokinin. Moreover, all the dominant lignans, including hinokinin, savinin, taiwanin A, taiwanin C, and helioxanthin were synthesized after savinin treatment for 24 h. In particular, synthesis of taiwanin A, which revealed heartwood formation in Taiwania.

In addition, the shrinkage of protoplast cells scattered in the culture medium after 100 μM savinin treatment (Figure 3b), indicated a slight disintegration of the cell membrane. As the savinin concentration increases to 150 μM, the cell membrane collapsed severely, and released complete vacuoles or organelles (Figure 3c), consistent with the characteristics of necrotic cell death (NCD). Therefore, the accumulation of savinin may cause ray parenchyma cells to undergo NCD, which is usually the case in plants. NCD is mainly caused by microbial infection, which echoes the previous study on the formation of heartwood that parenchyma PCD belongs to a biological factor−induced type [39]. Olvera−Carrillo et al. classified biotic−induced cell death as being mostly caused by oxidative stress, as indicated in their report [28]. Pathogen infections are known to induce oxidative stress. While heartwood formation can produce a large amount of lignans, which have been reported to possess antibacterial bioactivity, it is believed that pathogen infection is responsible for the PCD in xylem parenchyma cells. However, our previous study has demonstrated that genetics plays a major role in the regulation of annual rings in the sapwood of Taiwania [21]. If pathogen infection were the cause of heartwood formation, the number of annual rings would not be consistent. Therefore, there must be a genetic factor that induces oxidative stress, leading to PCD in xylem parenchyma cells. Furthermore, our study showed that the abundance of lignans in heartwood formation is primarily regulated by genetics [21]. Based on this evidence, we believe that lignan is the trigger for PCD in ray parenchyma cells in Taiwania.

To whether know savinin could induce oxidative stress. Figure 4 shows the results of detecting oxidative stress by H_2_DCF−DA. H_2_DCF−DA itself has no fluorescence and can freely enter and exit the cell membrane. However, after entering the cell, it will be hydrolyzed into 2′,7′-dichlorodihydrofluorescin (DCFH) by intracellular esterase. The deacetylated DCFH cannot penetrate the cell membrane, and will be oxidized by intracellular ROS to fluoresce 2′,7′-dichlorofluorescin (DCF). Therefore, this system can be used to detect the ROS content in protoplasts. As shown in Figure 4, it is evident that the levels of oxidative stress increased with with rising concentrations of savinin treatments. Notably, at 100 μM, the levels were the highest, representing an approximately 50% increase compared to the control, and 1000 μM may be that the protoplast is instantly destroyed, so its oxidative stress is comparable to that of the control group (from Figure 3c, it can be seen that in protoplasts were destroyed instantaneously at 150 μM savinin). However, the oxidative stress of hinokinin and taiwanin A were no induced. These results further suggest the accumulation of savinin may cause oxidative stress during PCD in ray parenchyma cells.

## 3. Materials and Methods

### 3.1. Plant Materials and Lignans

Wood cores were gathered in August 2014 from three 30−year−old Taiwania trees at the Huisun Experimental Forest Station of National Chung−Hsing University, with collection taking place at a height of 130 cm above the ground.; and was identified by Prof Yen−Hsueh Tseng, Department of Forestry, National Chung Hsing University. The voucher specimen was deposited in the herbarium of the same university. Wood cores were collected with an increment borer, and cut the tangential sections every 0.5 cm, which were 1.0 × 1.0 cm^2^ and thickness Less than 0.3 cm, the sample numbered 1–5 from sapwood to heartwood. Samples 1–4 were sapwood, and sample 5 was the part where the heartwood was first formed. 3−year−old Taiwania seedlings were for extracting protoplast. The six lignans (hinokinin, savinin, taiwanin A, taiwanin C, taiwanin E and helioxanthin (Appendix A) were purified and quantified according to Tsao et al. [2].

### 3.2. Analysis for Types of Induced Programmed Cell Death

The types of induced programmed cell death (PCD) in Taiwania were analyzed based on Olvera−Carrillo and his coworkers’ method [28], who introduced PCD−related genes, including 25 developmentally induced PCD−related genes, 27 biological factor−induced PCD−related genes, 27 osmotic pressure−induced PCD−related genes, and 30 genotoxicity−induced PCD−related genes, a total of 109 induced PCD−related genes by 82 experiments (Appendix A). These gene sequences were then searched and compared to the transcriptome database of Taiwania established by Professor Chu’s group [22,28,40]. We got each gene expression (FPKM) in developing xylem (DX), sapwood (SW), and transition zone (TZ) from the database. The ratio of gene expression in DX to SW and SW to TZ were analyzed by formula log (FPKM of SW/FPKM of DX) and log (FPKM of TZ/FPKM of SW), respectively. Analyzed which induced type of PCD belongs.

### 3.3. Positioning Analysis of Lignans in the Tangential Section of Taiwania

In order to confirm whether the biosynthesis of lignans is related to the PCD of ray parenchyma cells, it is necessary to determine lignans are accumulated in ray parenchyma cells. Our study used the growth cone to take out the transverse wood core of the 30−year−old Taiwaia, and cut the wood core tangential section specimens at intervals of 0.5 cm from the sapwood. The wood was numbered 1–5 in sequence towards the heartwood. The 1–4 test material was located in the sapwood. The No. 5 sample was taken from the part where the heartwood was initially formed. The samples were freeze−dried and analyzed by ToF−SIMS (Time−of−Flight Secondary Ion Mass) spectrometer (TOF−SIMS V; ION−TOF, Munster, Germany). Positive and negative ion spectra within the *m*/*z* 0-1850 mass range were obtained using a 22 keV Au1^+^ beam operated at a current of 0.01 pA, with a pulse width of 10 ns (<1 ns after bunching), and 2 μm spot diameter. The primary ion beam was rastered over a 200 × 200 μm^2^ area to investigate the distribution of the relative intensities of the ions derived from the extractives. All square images obtained were divided into 256 × 256 pixels with an acquisition time of approximately 15 min each. A low−energy pulsed primary ion gun (28.0 eV) was used for surface charge compensation. The detected *m*/*z* of 217, 245, 351, 353, 355 and 365 corresponded to taiwanin C, helioxanthin, taiwanin A, savinin, hinokinin and taiwanin E, respectively.

### 3.4. Wood Core Staining of ROS

To detect the presence of ROS, using nitroblue tetrazolium (NBT) to become insoluble diformazan upon reduction. The method is modified from Sun et al. [41]. A new wood core (Appendix A) obtained at 130 cm above the ground in three 30−year−old Taiwania from the Huisun Experimental Forest Station of National Chung−Hsing University, which quickly placed in a solution containing 6 mM NBT (in 10 mM Na−Citrate buffer, pH6). The infiltrated wood core kept in the dark for 2 h at room temperature to allow the formation of insoluble dark blue diformazan. The treated wood core was immersed in ethanol for 30 min, and the pigments could be removed except the insoluble diformazan.

### 3.5. In Vitro Assay in Protoplast

Lin et al. method was referred to establish the protoplast system of Taiwania xylem for in vitro experiments [42]. The formulations of cell wall digestion enzyme solution (CWD solution), buffer solution (MMG solution), and cell culture solution (cell culture solution) are listed in Appendix A. All solutions were finally prepared with 0.6 M mannitol to reach the target osmolarity.

#### 3.5.1. Protoplast Extraction

100 3−year−old Taiwania seedlings were cut into around 6 cm, then removed the bark was put into a 50 mL centrifuge tube. Added 40 mL of CWD solution into the tube, soak for 6 h, then replace the bark−free stems into a new 50 mL centrifuge tube, add 40 mL of MMG solution, shake slightly for 1 min, filter with a cell mesh, and use 100× *g* centrifuge for 5 min, remove 35 mL of supernatant, add 5 mL of 20% sucrose solution, centrifuge at 100× *g* for 5 min, absorb the mist layer in the middle, wash twice with MMG solution, and finally replace with cell culture solution and calculate the density of protoplasts.

#### 3.5.2. Analysis of Lignans in In Vitro

The extracted protoplasts were divided into 5 groups, namely the control group, the calcium chloride treatment group, the hydrogen peroxide treatment group, the hinokinin treatment group, and the savinin treatment group. Each group was prepared with 0.5 mL cell solution with a density of 5 × 10^3^ cells, the treating time was 24 h for each treatment. The control group was treated with DMSO, and the calcium chloride treatment group, hydrogen peroxide treatment group, hinokinin treatment group, and savinin treatment group were treated according to the concentration gradient. Calcium chloride treatment: Protoplasts were treated with 20, 60, 100 and 120 mM CaCl_2_ for 2 h, and finally treated with 120 mM for 18 h; Hydrogen peroxide treatment: Protoplasts were treated with 1, 3, 7, and 10 mM H_2_O_2_ for 2 h, followed by 20 mM for 14 h. Finally, treating with 30 mM H_2_O_2_ for 2 h; Hinokinin and savinin treatment groups: The treatment conditions were as follows, 1, 2, 6, 12, 20, 40, 60 and 80 μM hinokinin or savinin were treated for 1 h; continued with 100 mM hinokinin or savinin for 14 h, and finally treated with 150 μM hinokinin or savinin for 2 h. The treated protoplasts were using 100× *g* centrifuge for 5 min to remove the supernatant, freeze−dried, extracted with methanol, and quantitatively analyzed for six lignans, i.e., hinokinin, savinin, taiwanin A, taiwanin C, taiwanin E, and helioxanthin by UHPLC−MS.

### 3.6. Oxidative Stress Analysis of In Vitro

The extracted protoplasts were divided into 8 groups with 2 mL of cell solution at a density of 1 × 10^4^ per group, which were control group, 1 μM savinin treatment group, 10 μM savinin treatment group, 100 μM savinin treatment group, 1000 μM savinin treatment group, 100 μM hinokinin treatment group, 1 μM taiwanin A treatment group, and 10 μM taiwanin A treatment group; the treating time were all 24 h. After treatment, 5 μL of 10 mM 2′,7′-Dichlorodihydrofluorescein diacetate (H_2_DCF−DA) was added to react in the dark for 30 min, then centrifuged at 1000× *g* for 5 min; and 200 μL of the supernatant was transferred to a black 96−well plate. The microanalyzer detects the fluorescence absorption of excitation light at 504 nm and emission light at 529 nm.

### 3.7. Statistical Analysis

Statistical comparisons of the results were performed using the one-way ANOVA. Any significant differences were denoted as * *p* < 0.05.

## 4. Conclusions

Based on the results of the biological information analysis, the transformation of cambium into sapwood during the differentiation of Taiwania primarily occurs through genotoxicity−induced programmed cell death (PCD), leading to the death of tracheids. As the sapwood progresses into the transition zone, the ray parenchyma cells undergo PCD induced by biological factors. It has been further reported that the generation of oxidative stress eventually triggers the occurrence of PCD. Moreover, the results from ToF−SIMS analysis indicate that the production of lignans in Taiwania takes place predominantly in the ray parenchyma cells. The progression of accumulation for savinin, taiwanin A, taiwanin C, and helioxanthin from sapwood to heartwood substantiates the notion that savinin stimulates the programmed cell death (PCD) of ray parenchyma cells. This process consequently leads to the substantial generation of taiwanin A, taiwanin C, and helioxanthin. Based on the findings of this study, it can be inferred that ray parenchyma cells in the sapwood of Taiwania accumulate savinin. Once the accumulation of savinin reaches a certain level in the transition zone, it induces oxidative stress and triggers the production of a large quantity of lignans, such as taiwanin A, ultimately leading to the death of ray parenchyma cells. Therefore, savinin could be considered the driving factor in the formation of Taiwania’s heartwood.

## Figures and Tables

**Figure 1 plants-12-03031-f001:**
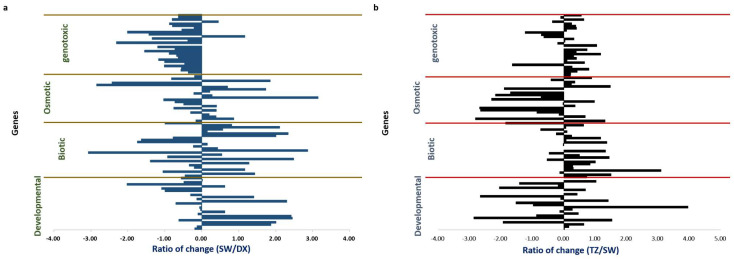
Gene expression difference in differentiating xylem (DX), sapwood (SW), and transition zone (TZ) from Taiwania (*Taiwania cryptomerioides*) base on the expression genes of programmed cell death in Arabidopsis thaliana published by Olvera−Carrillo et al. (2015) [28]. (**a**): The gene expression difference in the first step. The gene expression ration of SW/DX, and then the logarithmic conversion is performed. The positive value of this gene expression is higher in SW, and the negative value is higher in DX. (**b**): The gene expression difference in the second step. The gene expression ration of TZ/SW, and then the logarithmic conversion is performed. The positive value of this gene expression is higher in TZ, and the negative value is higher in DX.

**Figure 2 plants-12-03031-f002:**
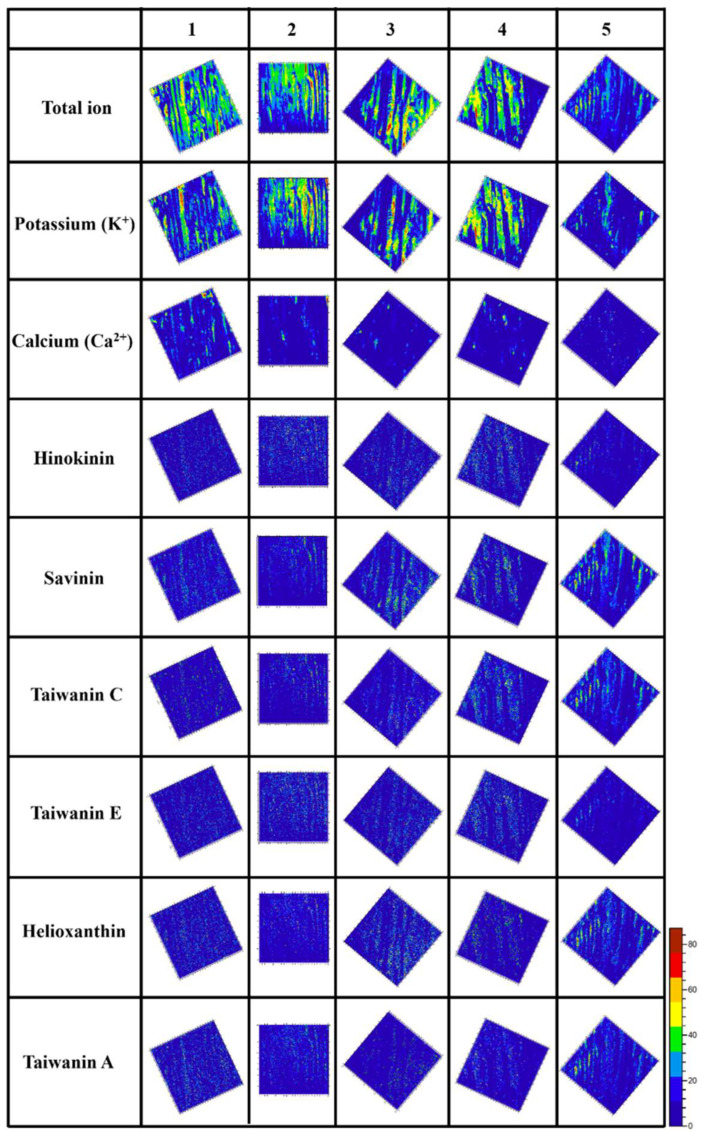
ToF−SIMS ion images of the tangential section of Taiwania (*Taiwania cryptomerioides*). Each row of ion images was each mapping target (included inorganic elements and lignans). Each column of ion images was the position of sapwood and heartwood. The No. 1–5 were that followed sapwood to heartwood. No. 1–4 were sapwood, No. 5 was heartwood. The information on dimension of analyzed area was 150 × 150 μm^2^.

**Figure 3 plants-12-03031-f003:**
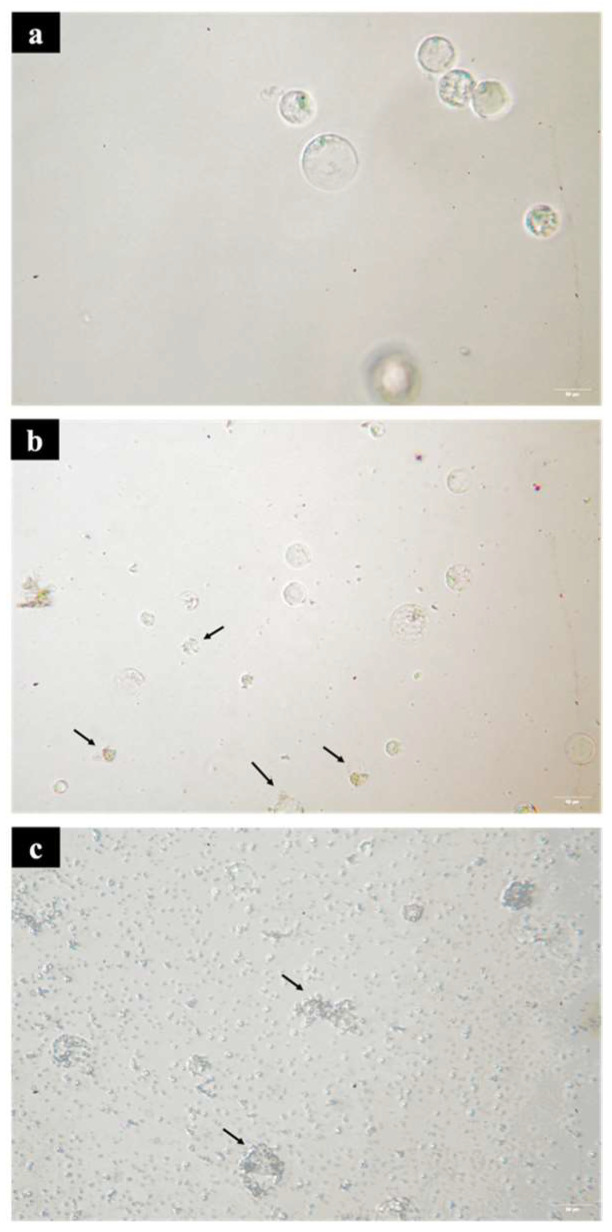
Optical microscopic observation of protoplasts after treating savinin. (**a**) without savinin treatment. (**b**) 100 μM savinin treatment, arrows indicated a slight disintegration of the cell membrane. (**c**) 150 μM savinin treatment, arrows indicated the cell membrane collapsed severely.

**Figure 4 plants-12-03031-f004:**
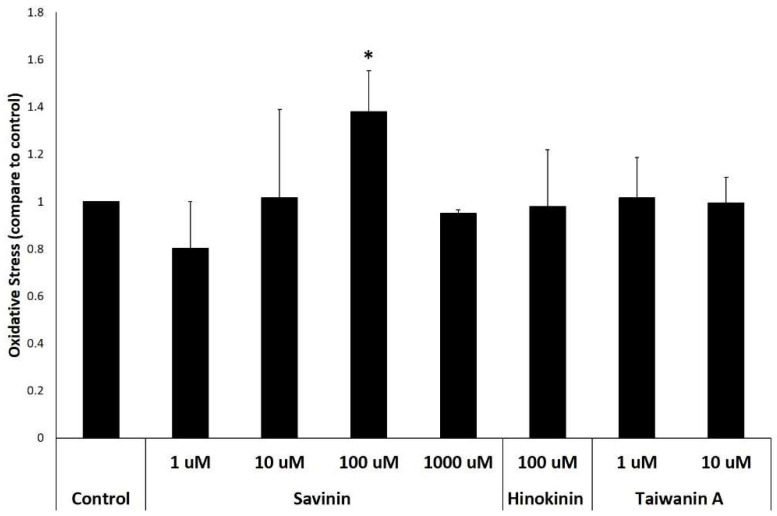
Analysis oxidative stress of protoplasts with different treatments after 24 h by H_2_DCF−DA. Error bars denote standard deviation. Asterisk denotes a significant difference from the control (*p* < 0.05).

**Table 1 plants-12-03031-t001:** Quantification of lignans synthesized by different treatment in the protoplast cells system.

	Concentration (μM)					
	Hinokinin	Savinin	Taiwanin A	Taiwanin C	Taiwanin E	Helioxanthin
Control	tr ^a^	tr	-	-	-	-
CaCl_2_ treatment	-	tr	-	-	-	-
H_2_O_2_ treatment	-	-	-	-	-	-
Hinokinin treatment	9.2	tr	-	0.6	-	1.2
Savinin treatment	tr	57.5	0.3	0.02	-	0.3

^a^ Trace: the concentration was less than 0.6 μM.

## Data Availability

All data generated or analyzed during this study are included in this published article and its Appendix A.

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
