# Peer review of "Savinin Triggers Programmed Cell Death of Ray Parenchyma Cells in Heartwood Formation of Taiwania cryptomerioides Hayata"

_plants, 2023, doi:10.3390/plants12173031_

Round 1

Reviewer 1 Report

The manuscript entitled “Savinin triggers programmed cell death of ray parenchyma cells in heartwood formation of Taiwania cryptomerioides Hayata” reported the relationship between lignans biosynthesis and 15 programmed cell death (PCD) of ray parenchyma cells during the heartwood formation of Taiwania. The study will have values for further understanding the regulation mechanism of PCD in parenchymal cells. However, the following should be addressed by authors before any decision could be made.

1.         Line 174, this sentence ‘we also confirmed the oxidative stress was higher in transition zone and heartwood’, Compared to what?

2.         Lines199-192, this sentence ‘no lignans were found during hydrogen peroxide treatment’ and ‘the accumulation of savinin may cause oxidative stress during PCD in ray parenchyma cells’ in line 236, it seems contradictious? Please make a reasonable explanation.

3.        In Fig.4, suggested to supplement difference on statistics, as well as the control.

4.         In method part, added statistical analysis for related data.

5.         In Conclusions section, suggested to rewrite the part, as possible as succinct, not repeated many results.

The paper should be corrected for English preferably by a native speaker.

Author Response

Comment 1:

Line 174, this sentence ‘we also confirmed the oxidative stress was higher in transition zone and heartwood’, Compared to what?

Response 1:

Thank you for bringing this to our attention. We have compared the transition zone and heartwood to sapwood and have made revision to the sentence in Lines 174-175 accordingly.

Comment 2:

Lines199-192, this sentence ‘no lignans were found during hydrogen peroxide treatment’ and ‘the accumulation of savinin may cause oxidative stress during PCD in ray parenchyma cells’ in line 236, it seems contradictious? Please make a reasonable explanation.

Response 2:

Thank you for your comment. Firstly, it is important to note that hydrogen peroxide itself is considered an oxidative stress agent, so it was already acting as an intermediate molecule. However, savinin served as the source for generating oxidative stress. Secondly, the group subjected to hydrogen peroxide treatment lacked the necessary substrate to undergo lignan formation. On the other hand, savinin was capable of inducing oxidative stress and transforming itself into other lignans.

Comment 3:

In Fig.4, suggested to supplement difference on statistics, as well as the control.

Response 3:

Thank you for your suggestion. We have revised it in Lines 238-240.

Comment 4:

In method part, added statistical analysis for related data.

Response 4:

Thank you for your kind reminding, we have added it in Lines 333-335.

Comment 5:

In Conclusions section, suggested to rewrite the part, as possible as succinct, not repeated many results.

Response 5:

Thank you for your suggestion. We have revised the conclusions in Lines 337-351.

Reviewer 2 Report

The study is carefully designed, implemented and reported.

This reviewer has difficulties in understanding a statement on page 7, regarding the relation of pathogen infection and annual ring formation.

For this reviewer, the co-ordinate system of Fig. 1 did not appear clear.

In several occasions, singular and plural expressions appear to be mixing. A few sentences appear to be missing words.

Author Response

Comment 1:

This reviewer has difficulties in understanding a statement on page 7, regarding the relation of pathogen infection and annual ring formation.

Response 1:

Thank you for your comment. Regarding page 7, we have provided an explanation for  why heartwood formation in Taiwania was classified as biotic-induced. In simpler terms, following the classification proposed by Olvera-Carrillo et al., heartwood formation in Taiwania was attributed biotic-induced programmed cell death. However, our previous study demonstrated that heartwood formation is regulated by genes, as supported by the heritability as annual rings in Taiwania (Tsao et al., 2019). This indicates a distinction from biological factor control. Moreover, literature suggests that pathogen infection can trigger oxidative stress (Olvera-Carrillo et al., 2015). Therefore, the presence of oxidative stress was the basis for classifying heartwood formation in Taiwania as biotic-induced programmed cell death.

Comment 2:

For this reviewer, the co-ordinate system of Fig. 1 did not appear clear.

Response 2:

Thank you for your kind reminder. Each bar in the figure represents a gene specific gene. For detailed information regarding the genes and corresponding numerical data from the transcriptome, please refer to table S1.

Reviewer 3 Report

The manuscript submitted for review, entitled ‘Savinin triggers programmed cell death of ray parenchyma cells in heartwood formation of Taiwania cryptomerioides Hayata’, presents an attempt to address the relationship between accumulation of savinin and programmed cell death. Key concept of the manuscript seems clear and valuable. The paper presents a logical sequence of studies following one another and is structured. The authors refer to previous studies to provide reasons for the next step.

My objections concern some details and methodological shortcuts. As it currently stands, some parts of the study are impossible to replicate or verify due to too little detail in some places. Comments are included below.

In order to study the process of heartwood formation, it is first necessary to describe the development of this process in Taiwaniana. At what age does the tree start the process of heartwood formation? In the study, the heartwood part was considered the 5th annual ring. Is this a normal for this species? With pines heartwood can occur after several or even 30 years, variability is therefore very high. It is necessary to refer to the average age of the heartwood formation of Taiwaniana species.

The cited publication (1) does not address the relationship of heartwood to the properties mentioned. In particular, the relationship of heartwood to strength in comparison to sapwood is not clear, since it depends mainly on cellulose, not extractives. Mechanical properties in various species are similar in sapwood and heartwood.

Selection of materials for testing.

Practically, no details are provided on the selection of material for the study.

l. 243 A 30-year-old Taiwania... How many samples were taken? From how many trees? From what trunk height were the cores taken? If the results presented in the paper (e.g., in figure 2) are only from a small sample (one core, one tree) then they do not provide mathematical proof, but only an assumption. Taiwaniana is a rare and valuable species and it may be difficult to obtain suitable material, but nothing was mentioned.

Figure 3 – photographs are of poor quality, too dark, key elements should be indicated and described in the photo

ll. 74-76 it was mentioned that “Taiwania is the appropriate material for studying heartwood formation, due to it can easily distinguish the heart wood, sapwood wood, and transition zone”… However, in the study there is a separation only between sapwood and heartwood (methods 3.1, Fig. 2). The authors mention that they were involved in transition zone studies (l. 128), so they have experience in this area. Was the transition zone present in the studied wood or not?

Please provide a clear photographs or scans of the raw wood with an indication of the zones (sapwood, heartwood, transition zone if exist) and the sampling location on the tree. In the absence of such photographs, a diagram should be provided as a drawing. The fragments in figure S4 are illegible, and it is also not clear from which part of the trunk the tested samples were taken.

Figures composed of multiple diagrams, e.g., S2, should be described with letters (e.g., a,b,c, etc.). Figures lack statistical significance of differences - this is especially true for the first two of Fig. S2 (K+, Ca2+) with similar values. Axis descriptions are poorly visible (small font).

The statistical tests in the paper are poor because not even descriptive statistics are presented. Tests-t are used to compare two groups, but what about the other groups against each other?

The results shown in Figure 4 are not clear because the comparisons are not fully presented. Why are 4 treatment shown for savinin, but only one for hinokinin and two for taiwanin A? According to scientific rules all tested treatments should be visible or please give reason why you show selected treatments. These results also require detailed statistical support.

Please correct the headings of tables S3 and S4 (only “ml” is given in solutions, no “g”).

Accumulation of savinin has been found to be the main factor initiating the formation of heartwood, but the tests were performed only in vitro therefore please consider if this conclusion should be stated rather as presumption.

Overall, the work brings some novelty, so the results should be better presented and statistically supported.

Author Response

Comment 1:
In order to study the process of heartwood formation, it is first necessary to describe the development of this process in Taiwaniana. At what age does the tree start the process of heartwood formation? In the study, the heartwood part was considered the 5th annual ring. Is this a normal for this species? With pines heartwood can occur after several or even 30 years, variability is therefore very high. It is necessary to refer to the average age of the heartwood formation of Taiwaniana species.
Response 1:
Thank you for your comment. According to our prior research (Tsao et al., 2016), Taiwania initiates the formation of heartwood at approximately 10-13 years of age. Please note that the term "5th" does not pertain to annual rings; rather, it denotes the sample labeled as No. 5, extracted from a 30-year-old Taiwania tree. We have rectified the use of "5th" to "No. 5" in Line 275.

Tsao, N.W.; Sun, Y.H.; Chien, S.C.; Chu, F.H.; Chang, S.T.; Kuo, Y.H.; Wang, S.-Y. Content and distribution of lignans in Taiwania cryptomerioides Hayata. 2016, 70, 511-518, doi:10.1515/hf-2015-0154.

Comment 2:
The cited publication (1) does not address the relationship of heartwood to the properties mentioned. In particular, the relationship of heartwood to strength in comparison to sapwood is not clear, since it depends mainly on cellulose, not extractives. Mechanical properties in various species are similar in sapwood and heartwood.
Response 2:
Thank you for your suggestion. We have deleted the strength in Line 39.

Comment 3:
Selection of materials for testing.
Practically, no details are provided on the selection of material for the study.
l. 243 A 30-year-old Taiwania... How many samples were taken? From how many trees? From what trunk height were the cores taken? If the results presented in the paper (e.g., in figure 2) are only from a small sample (one core, one tree) then they do not provide mathematical proof, but only an assumption. Taiwaniana is a rare and valuable species and it may be difficult to obtain suitable material, but nothing was mentioned.
Response 3:
Thank you for your suggestion. We have revised it in Line 246-248. “Wood cores were gathered in August 2014 from three 30-year-old Taiwania trees at the Huisun Experimental Forest Station of National Chung-Hsing University, with collection taking place at a height of 130 cm above the ground.”

Comment 4:
Figure 3 – photographs are of poor quality, too dark, key elements should be indicated and described in the photo
Response 4:
Thank you for your thoughtful suggestion. We have made revisions to Figure 3 as indicated in Lines 223-224.

Comment 5:
ll. 74-76 it was mentioned that “Taiwania is the appropriate material for studying heartwood formation, due to it can easily distinguish the heart wood, sapwood wood, and transition zone”… However, in the study there is a separation only between sapwood and heartwood (methods 3.1, Fig. 2). The authors mention that they were involved in transition zone studies (l. 128), so they have experience in this area. Was the transition zone present in the studied wood or not?
Response 5:
Thank you for your comment. In our prior research, we did indeed encompass the transition zone, albeit within a limited scope. However, in this current study, owing to challenges associated with acquiring tangential sections of the transition zone, our research exclusively concentrates on the investigation of sapwood and heartwood.

Comment 6:
Please provide a clear photographs or scans of the raw wood with an indication of the zones (sapwood, heartwood, transition zone if exist) and the sampling location on the tree. In the absence of such photographs, a diagram should be provided as a drawing. The fragments in figure S4 are illegible, and it is also not clear from which part of the trunk the tested samples were taken.
Response 6:
Thank you for your suggestion. We have added information about figure S4 sample in Line 290.

Comment 7:
Figures composed of multiple diagrams, e.g., S2, should be described with letters (e.g., a,b,c, etc.). Figures lack statistical significance of differences - this is especially true for the first two of Fig. S2 (K+, Ca2+) with similar values. Axis descriptions are poorly visible (small font).
Response 7:
Thank you for your valuable suggestion. As this constitutes a sample dataset, substantial statistical analysis is not applicable. Nevertheless, the discernible trend in lignan alterations remains observable. Consequently, we have made revisions to Figure S2.

Comment 8:
The statistical tests in the paper are poor because not even descriptive statistics are presented. Tests-t are used to compare two groups, but what about the other groups against each other?
Response 8:
Thank you for sharing your comment. There was an error in the notation of the t-test method, which has now been rectified to a one-way ANOVA, as reflected in Line 339.

Comment 9:
The results shown in Figure 4 are not clear because the comparisons are not fully presented. Why are 4 treatment shown for savinin, but only one for hinokinin and two for taiwanin A? According to scientific rules all tested treatments should be visible or please give reason why you show selected treatments. These results also require detailed statistical support.
Response 9:
Thank you for your insightful comment. Our study's findings underscore the pivotal role of oxidative stress in the process of heartwood formation. Furthermore, within this process, savinin emerges as a significant contributor, with hinokinin assumedly acting as an antecedent to savinin. Notably, Taiwanin A, an exclusive compound indigenous to Taiwania, has been suggested in literature to possess the capacity to induce oxidative stress in cancer cells. Thus, these three compounds were chosen for experimentation. The selection of concentrations stems from protoplast tests, where an observation was made that a concentration of 100 mM serves as a catalyst for the transformation of hinokinin and savinin into subsequent products. Given the integral role of savinin in triggering heartwood formation within Taiwania, concentrations were specifically varied for it. In the case of taiwanin A, its concentration was determined with reference to prior cancer cell experiments documented in existing literature.

Comment 10:
Please correct the headings of tables S3 and S4 (only “ml” is given in solutions, no “g”).
Response 10:
Thank you for pointing this out. We have deleted the weight (g) in table S3 and S4.

Comment 11:
Accumulation of savinin has been found to be the main factor initiating the formation of heartwood, but the tests were performed only in vitro therefore please consider if this conclusion should be stated rather as presumption.
Overall, the work brings some novelty, so the results should be better presented and statistically supported.
Response 11:
Thank you for your suggestion. We have revised it in Lines 348-351. “The progression of accumulation for savinin, taiwanin A, taiwanin C, and helioxanthin from sapwood to heartwood substantiates the notion that savinin stimulates the programmed cell death (PCD) of ray parenchyma cells. This process consequently leads to the substantial generation of taiwanin A, taiwanin C, and helioxanthin.”.

Reviewer 4 Report

Dear Authors and Editors,

heartwood formation is still an interesting and exciting part of biochemistry and wood research opening new results and insights into the process with the constant development of analytical techniques. The present contribution is a good example to this. The multi-method study of the heartwood formation on Taiwania provides results into the process of the triggers and causes of heartwood formation which will provide data to researchers dealing with the same topic and with other tree species for comparison.

In short I found the work valuable and acceptable to Plants, nevertheless not in the present form, but after revision.

My remarks in detail:

The most apparent thing to correct is English language. The manuscript is full of typos and examples of bad language use, thus it should be throughoutly checked by a native English speaker professional, which is curucial.

Some (!) examples: L15 lignan not lignans

L70: Taiwania NOT Taiwnaia

L69: lignan not lignans

L128, 130, 169, 285, 319: bad language use

L246-248: formatting

Supplementary material: Caption of Figure S2, "That dividing the intesity..." What does this sentence mean?

Figures: Figure2 is only readably in pdf format not printed. Numbers cannot be recognized. Please improve

Figure 3: Only interpreatble in pdf format, it would do good to put some more contrast to the figure so that differences would be better readably in print and also on screen.

Figure 4: what do error bars denote here? Please indicate in caption! (standard deviation, confidence range etc?)

Tables: TableS1: what do FPKM(DX) etc mean here? Please describe in the caption. What is the unit of these values please inducate!

Table S3, S4: only voulume (mL) indicated in table. no weight (g) indicated in tables so please omit!

Article contains 40 cited references, which is OK but somewhat low to im my opinion. What is lacking to me is the citation of the latest results on the chemical aspects of heartwood formation. In fact the references at L59: [9-16] range from from 2002-2014, which I found quite oldish, especially in the light of the fact that quite much has been done in the chemistry on hartwood formation, enabled by latest techniques. Latest results should be cited at L59 or somewhere else in the article:

Hofmann, T.; Guran, R.; Zitka, O.; Visi-Rajczi, E.; Albert, L. Liquid Chromatographic/Mass Spectrometric Study on the Role of Beech (Fagus sylvatica L.) Wood Polyphenols in Red Heartwood Formation. Forests 202213, 10. https://doi.org/10.3390/f13010010

Felhofer M, Prats-Mateu B, Bock P, Gierlinger N. Antifungal stilbene impregnation: transport and distribution on the micron-level. Tree Physiol. 2018 Oct 1;38(10):1526-1537. doi: 10.1093/treephys/tpy073. PMID: 29992254; PMCID: PMC6198867.

Very poor style. Should be improved significantly. Also many typos.

Author Response

Comment 1:
The most apparent thing to correct is English language. The manuscript is full of typos and examples of bad language use, thus it should be throughoutly checked by a native English speaker professional, which is curucial.
Response 1:
Thank you for your comment. We have asked native English speaker to edit the manuscript.

Comment 2:
Some (!) examples: L15 lignan not lignans
Response 2:
Thank you for pointing this out. We have corrected it in Line 15.

Comment 3:
L70: Taiwania NOT Taiwnaia
Response 3:
Your observation is much appreciated. The necessary correction has been made, as indicated in Line 70.

Comment 4:
L69: lignan not lignans
Response 4:
We are grateful for bringing this to our attention. The identified issue has been rectified, as reflected in Line 69.

Comment 5:
L128, 130, 169, 285, 319: bad language use
Response 5:
Thank you for your suggestion. We have improved these sentences in Lines 127-133, 160-183, 287-288, 323and 325.

Comment 6:
L246-248: formatting
Response 6:
Thank you for pointing this out. We have corrected it in Line 249-255.

Comment 7:
Supplementary material: Caption of Figure S2, "That dividing the intensity..." What does this sentence mean?
Response 7:
Thank you for your comment. It a normalization method, we have revised the sentence in figure S2 legend.

Comment 8:
Figures: Figure 2 is only readably in pdf format not printed. Numbers cannot be recognized. Please improve
Response 8:
Thank you for your comment. We have revised figure 2.

Comment 9:
Figure 3: Only interpreatble in pdf format, it would do good to put some more contrast to the figure so that differences would be better readably in print and also on screen.
Response 9:
Thank you for your comment. We have revised figure 3.

Comment 10:
Figure 4: what do error bars denote here? Please indicate in caption! (standard deviation, confidence range etc?)
Response 10:
Thank you for your suggestion. Error bars denote standard deviation. We have revised it in Line 245.

Comment 11:
Tables: TableS1: what do FPKM(DX) etc mean here? Please describe in the caption. What is the unit of these values please inducate!
Response 11:
We greatly appreciate your comment. In response, we have incorporated annotations into Table S1.

Comment 12:
Table S3, S4: only voulume (mL) indicated in table. no weight (g) indicated in tables so please omit!
Response 12:
We sincerely thank you for bringing this to our attention. As per your observation, we have now removed the weight (g) entries from tables S3 and S4.

Comment 13:
Article contains 40 cited references, which is OK but somewhat low to im my opinion. What is lacking to me is the citation of the latest results on the chemical aspects of heartwood formation. In fact, the references at L59: [9-16] range from from 2002-2014, which I found quite oldish, especially in the light of the fact that quite much has been done in the chemistry on hartwood formation, enabled by latest techniques. Latest results should be cited at L59 or somewhere else in the article:
Hofmann, T.; Guran, R.; Zitka, O.; Visi-Rajczi, E.; Albert, L. Liquid Chromatographic/Mass Spectrometric Study on the Role of Beech (Fagus sylvatica L.) Wood Polyphenols in Red Heartwood Formation. Forests 2022, 13, 10. https://doi.org/10.3390/f13010010
Felhofer M, Prats-Mateu B, Bock P, Gierlinger N. Antifungal stilbene impregnation: transport and distribution on the micron-level. Tree Physiol. 2018 Oct 1;38(10):1526-1537. doi: 10.1093/treephys/tpy073. PMID: 29992254; PMCID: PMC6198867.
Response 13:
We appreciate your valuable suggestion. In response, we have included references 17 and 18 as specified (Lines 407-410).

Round 2

Reviewer 3 Report

The Authors made important corrections to the manuscript, making it more readable. Particularly important changes were made in crucial places, such as the quality of photographs their description and in key sentences into the text of manuscripts. Other issues have been sufficiently clarified. I believe that the manuscript is ready. I thank the Authors for their cooperation and wish them successful proceedings in the next step of the editorial process.

Reviewer 4 Report

Dear Authors and Editors,

the authors have made significant changes to the manuscript, figures were updated and ambigious paragraphs were changes. Also references were added and figure captions edited as suggested. I think the work can be published how. It is a nice contribution to the understanding of heartwood formation in trees, from which future researches will benefit.

Maybe a final language check would be good if the authors have not done so yet.